# Environment and weight class linked to skin microbiome structure of juvenile Eastern hellbenders (*Cryptobranchus alleganiensis alleganiensis*) in human care

Andrea C. Aplasca[1]*, Peter B. Johantgen[2], Christopher Madden[1], Kilmer Soares[1], Randall E. Junge[3], Vanessa L. Hale[1,4☯]*, Mark Flint[1,5☯]

1 Department of Veterinary Preventive Medicine, The Ohio State University College of Veterinary Medicine, Columbus, Ohio, United States of America, 2 Shores Department, Columbus Zoo and Aquarium, Powell, Ohio, United States of America, 3 Department of Animal Health, Columbus Zoo and Aquarium, Powell, Ohio, United States of America, 4 Center of Microbiome Science, The Ohio State University, Columbus, Ohio, United States of America, 5 One Welfare and Sustainability Center, The Ohio State University College of Veterinary Medicine, Columbus, Ohio, United States of America

☯ These authors contributed equally to this work.
* aplasca@gmail.com (ACA); hale.502@osu.edu (VLH)

## Abstract

Amphibian skin is integral to promoting normal physiological processes in the body and promotes both innate and adaptive immunity against pathogens. The amphibian skin microbiota is comprised of a complex assemblage of microbes and is shaped by internal host characteristics and external influences. Skin disease is a significant source of morbidity and mortality in amphibians, and increasing research has shown that the amphibian skin microbiota is an important component in host health. The Eastern hellbender (*Cryptobranchus alleganiensis alleganiensis*) is a giant salamander declining in many parts of its range, and captive-rearing programs are important to hellbender recovery efforts. Survival rates of juvenile hellbenders in captive-rearing programs are highly variable, and mortality rates are overall poorly understood. Deceased juvenile hellbenders often present with low body condition and skin abnormalities. To investigate potential links between the skin microbiota and body condition, we collected skin swab samples from 116 juvenile hellbenders and water samples from two holding tanks in a captive-rearing program. We used 16s rRNA gene sequencing to characterize the skin and water microbiota and observed significant differences in the skin microbiota by weight class and tank. The skin microbiota of hellbenders that were housed in tanks in close proximity were generally more similar than those housed physically distant. A single taxa, *Parcubacteria*, was differentially abundant by weight class only and observed in higher abundance in low weight hellbenders. These results suggest a specific association between this taxa and Low weight hellbenders. Additional research is needed to investigate how husbandry

**Data availability statement:** All sequencing data files are available from the NCBI database (accession number: BioProject ID: PRJNA722246).

**Funding:** Support for this project was provided by the Ohio State University Powers One Health Residency Fund (ACA), the Ohio State University Department of Veterinary Preventive Medicine (ACA, CM, KS, VLH, MF), and the Columbus Zoo and Aquarium (PBJ, REJ). The funders had no role in the study design, data collection and analysis, decision to publish, or preparation of the manuscript.

**Competing interests:** The authors have declared that no competing interests exist.

factors and potential pathogenic organisms, such as *Parcubacteria*, impact the skin microbiota of hellbenders and ultimately morbidity and mortality in the species.

## Introduction

Amphibian population declines have been documented over the past several decades across a diverse range of geographic regions and are linked to historic overharvesting, habitat loss and degradation, environmental contamination, climate change, and disease such as chytridiomycosis [1–5]. Of approximately 8500 described amphibian species, the International Union for the Conservation of Nature estimates 40% are threatened [6]. Amphibian skin is integral to promoting normal physiologic processes in the body and promotes both innate and adaptive immunity against a variety of pathogens [7–9]. Skin diseases can therefore be a significant source of morbidity and mortality in captive and free-ranging amphibian populations [10]. Skin disease may occur following primary exposure to a pathogenic organism, however skin infections have also been linked to improper husbandry, environmental toxins, and inbreeding depression [9]. Recent research has also highlighted the role of the microbiota in host health [11]. An organism's habitat, including the soil, water, plants, and other animals to which they may be exposed, are important influences on a host. Broad scale biogeography and climate factors like precipitation and temperature patterns can affect host body temperature, skin shedding, physiologic processes, behavior, and environmental conditions that all interact in complex ways to shape the skin microbiome [12].

Advances in characterizing and understanding various factors that influence the amphibian skin microbiota can help improve the management for amphibians under human care as well as those in free-ranging settings. Experimental studies evaluating the application of probiotic therapies to amphibian hosts have demonstrated positive health effects in response to disease [7–8]. Additionally, serial sampling of the skin microbiota of wild amphibians and those under human care can reveal changes in skin bacterial diversity over time. In some amphibian populations, depletion of skin bacterial diversity has been related to decreased normal skin defenses and increased susceptibility to skin pathogens [8]. Because amphibian skin is so integral to whole body systemic health, improved understanding of the skin microbiota can help advance the health and management of amphibians on both an individual and population level.

The Eastern hellbender *(Cryptobranchus alleganiensis alleganiensis)* is a giant salamander of the family Cryptobranchidae and their native habitat spans several river systems across parts of the Northeast, Mid-Atlantic, and Midwest United States [13]. The cryptic nature of hellbenders makes accurate population estimates challenging, however, field surveys report the absence of hellbenders in waterways in which they have historically been found and low numbers of larvae and juveniles relative to adults [14]. The recovery of hellbenders is an important objective in many parts of its range [15]. In the state of Ohio, USA, the Columbus Zoo and Aquarium (CZA) partners with local wildlife authorities to collect hellbender eggs from the wild, captive-rear juvenile hellbenders, and release hellbenders to their native sites at approximately two to three years old [14]. In the past 15 years, the CZA has

successfully reintroduced over 1000 juvenile Eastern hellbenders to their native ranges, however historic survival rates during this three-year captive-rearing period have varied widely (20–85%) amongst different cohorts. Deceased juvenile hellbenders often present with low body condition and cutaneous lesions, including ulceration and necrosis; however, causes for morbidity and mortality are overall poorly understood [16].

Based on these findings we sought to investigate the skin microbiota of hellbenders by (1) characterizing the skin microbiota of hellbenders under human care with grossly normal skin in the CZA Eastern hellbender captive-rearing program and (2) analyzing the skin microbial diversity and composition of hellbenders by weight class and tank to assess for microbial associations with body condition and environment.

## Materials and methods

### Animals and husbandry

The Columbus Zoo and Aquarium (CZA) maintains a large captive-rearing program for hellbenders where approximately 300–600 animals from different cohorts are held annually. Each year the Ohio Hellbender Partnership collects eggs from the wild with the approval of the Ohio Department of Natural Resources Division of Wildlife. The CZA cultivates a subset of the eggs collected and rears hellbenders for approximately three years after which juvenile individuals are released to their native free-ranging sites in Ohio. In September 2017, the CZA received eggs collected from a single location and clutch during a single collection event from an undisclosed site in Ohio. Young hellbenders were captive-reared until the time of sampling at approximately two and a half years old (June 2020).

Animals were maintained in long-term isolation rooms with biosecurity and husbandry practices instituted to reduce the potential for disease transmission to and within the captive-reared group. Animals were serviced by dedicated staff, dedicated footwear and clean clothing were worn by all staff entering isolation rooms, footbaths containing Virkon™ 1% solution (The Chemours Company LLC, Wilmington, Delaware) were utilized before entering isolation rooms, dedicated tools and equipment were used in each isolation room, and all staff washed hands before or upon entering isolation rooms with soap and water. Animals were group-housed in multiple tanks and animals of similar body weight and size were generally grouped together (Fig 1 and Table 1). The majority of hellbenders were housed in fiberglass tanks (152 x 76 x 61 cm) and a small proportion of individuals (approximately 10% of group) were housed in glass aquariums (189-liter capacity measuring 91 x 46 x 46 cm and 284-liter capacity measuring 122 x 46 x 51 cm). Water for all tanks was supplied from public drinking water supplier, Del-Co. Water was aged and aerated for a minimum of 48 hours to facilitate dechlorination and temperature equilibration to appropriate ambient room temperature. Each fiberglass tank contained approximately 265 liters of water. A separate sump holding approximately 227 liters of water with 25 μm filter socks for mechanical filtration and Bio-Balls for biological filtration was used for water filtration for all tanks. Each sump filtration system was connected to two fiberglass tanks. (Tanks AQ-R2-2B and MF-2B were connected to other fiberglass tanks not included in this study.) Each glass aquarium contained undergravel filters to provide mechanical and biological filtration. Partial water changes (approximately 90%) were performed weekly.

Water quality parameters were maintained at the following levels: water temperature between 66.0–73.0 degrees Fahrenheit, pH between 7.4–7.6, dissolved oxygen between 98–100%, specific conductivity between 250–500 μS/cm, nitrites and ammonia at 0 mg/l, and nitrates at less than 5.0 mg/l. Water temperature was maintained at room temperature in climate-controlled rooms and measured daily. pH, conductivity, and dissolved oxygen were measured weekly. Nitrates, ammonia, and nitrates were measured monthly. Each fiberglass tank contained between 21–36 individuals. Each 189-liter glass aquarium contained two or three individuals and each 284-liter glass aquarium contained six individuals. Each holding area contained multiple hides made of terracotta or limestone to provide cover to animals. No substrate was present in tanks in order to minimize accumulation of leftover food and animal disturbance during routine husbandry, and to optimize water quality overall.

Animals were all broadcast fed a compositionally similar diet three times per week totaling 2–4% of estimated animal body mass per tank. The diet was comprised of a mix of fresh black worms, frozen and thawed freshwater mysis shrimp

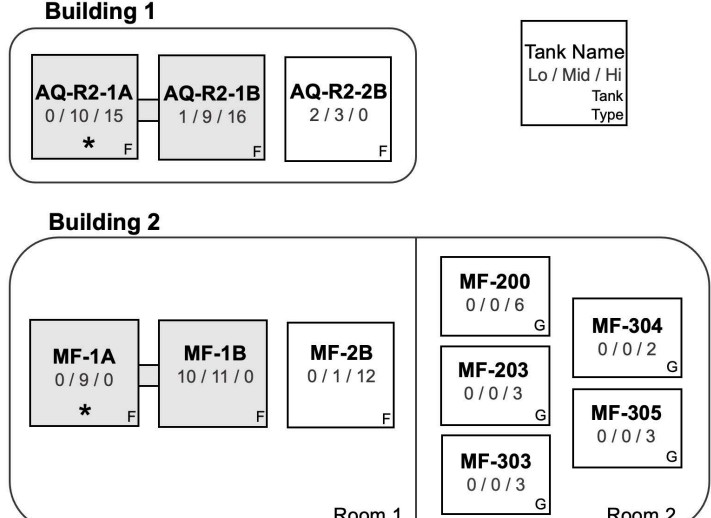

**Fig 1. Distribution of sampled animals by tank and weight class.** Animals were housed in multiple tanks in multiple buildings. The number of Low weight, Mid weight, and High weight animals housed in each tank is indicated below the tank name. Tanks that are shaded grey and connected by a grey bar had a shared water and filtration system. Fiberglass tanks are marked (F) and glass aquaria are marked (G). Water samples were collected from tanks marked with an asterisk (*).

**Table 1. Distribution of sampled animals by tank and weight class.**

| Tank | Number of sampled animals by weight class (Number of animals included in analysis) | | |
|---|---|---|---|
| | Low | Mid | High |
| AQ-R2-1A[1] | – | 10 (4) | 15 (11) |
| AQ-R2-1B[1] | 1 (1) | 9 (6) | 16 (13) |
| AQ-R2-2B | 2 (2) | 3 (3) | – |
| MF-1A[2] | – | 9 (9) | – |
| MF-1B[2] | 10 (9) | 11 (9) | – |
| MF-2B | – | 1 (1) | 12 (10) |
| MF-200 | – | – | 6 (4) |
| MF-203 | – | – | 3 (1) |
| MF-303 | – | – | 3 (2) |
| MF-304 | – | – | 2 (2) |
| MF-305 | – | – | 3 (3) |
| **TOTAL** | 13 (12) | 43 (32) | 60 (46) |

Tank superscripts note tanks with a shared water and filtration system.
Samples with fewer than 1800 reads were removed from analyses.

(*Mysis* spp.), frozen and thawed krill (order Euphausiacea), frozen and thawed lake smelt (family Osmeridae), live earthworms (*Lumbricus terrestris*), and occasional live superworms (*Zophobas morio*).

Animals were classified into three weight classes: Low weight (≤ 50 g), Mid weight (50–80 g), and High weight (≥ 80 g). The weight ranges for each weight class were determined based on the clinical history of this specific cohort of

hellbenders, historic institutional data, and the scaled mass index (SMI) to estimate 3 categories of body condition. Specifically, hellbenders from this cohort that died approximately 6 months prior to the current study often exhibited low body weight (< 50 g) and cutaneous lesions [16]). Additionally, similarly aged animals from previous cohorts (33–34 mos old, n = 132) had a mean body weight of 83.9 g and a mean total length of 26.5 cm. The SMI was calculated as body weight x (20/total body length)$^{2.98}$ based on institutional data from multiple previous hellbender cohorts [17]. Thus, the weight ranges and weight classes as defined above were established to categorize three distinct categories of weight with clinical relevance to both this specific study cohort and previous cohorts.

### Skin swab sample collection

Skin swab samples from 116 hellbenders (approximately 33% of the cohort) were collected as follows. Animals were removed from their normal holding tanks by hand or with the aid of small mesh dip-nets and placed in a sanitized 3.8 liter plastic holding tank in preparation for skin swab sample collection. Animals were rinsed by gently by pouring 100 ml of sterile water over the dorsal aspect of the body surface and skin swabs were collected using sterile polyester-tipped swabs (Puritan Medical Products Company LLC, Guilford, Maine, modified from Hernandez-Gomez et al. [18]). Each animal was swabbed five times on the mid-dorsal aspect of the body. Swabs were stored in sterile 2-ml microcentrifuge tubes, placed on frozen ice packs, and moved to a −80°C freezer within 2 hours for storage until DNA extraction. All animals had grossly normal skin and no skin abnormalities were noted in any animal at the time of sampling. Animals were returned to their holding tanks following sample collection. The plastic holding tank was cleaned between each animal. Animals were monitored to ensure that the dorsal aspect of the body was not in physical contact with the plastic holding tank at any time.

### Tank water sample collection

Water samples were collected from two tanks: AQ-R2-1A and MF-1A. These tanks were selected to capture samples from the two largest water systems which housed the largest number of animals (Fig 1). Isopropyl alcohol was used to rinse the external surface of a 50-mL conical tube and the tube was air dried completely prior to sample collection. The dry conical tube was then gently submerged into the tank at mid-depth, filled completely, and immediately closed. Water samples were placed on frozen ice packs and transferred to the laboratory for DNA extraction.

### DNA extraction, amplification, and sequencing

DNA extraction on all hellbender skin swabs was completed using QIAamp PowerFecal DNA Kits (Qiagen, Venlo, Netherlands). Water samples (50 mL) were concentrated by filtering through a 0.2μm filter (Thermo Scientific, Waltham, MA, USA) to trap bacteria. The filters then underwent DNA extraction using the QIAamp Powerfecal Pro DNA Kit (Qiagen, Venlo, Netherlands). DNA extraction was completed on three blank (no sample) tubes to serve as negative controls (one blank per extraction kit used). DNA concentrations were measured with a Qubit Fluorometer 4 (Invitrogen, Carlsbad, CA, USA). DNA purity was measured with a NanoDrop 1000 Spectrophotometer (Thermo Scientific, Waltham, MA, USA). DNA was submitted to Argonne National Laboratory for library preparation and 16S rRNA gene sequencing (V4 region) on an Illumina MiSeq using Earth Microbiome Project primers 515F and 806R [19,20]. All sequencing data files are available from the NCBI database (accession number: BioProject ID: PRJNA722246).

### 16S rRNA sequence processing and analysis

DNA from a total of 93 skin swab samples, 2 water samples, and 3 negative controls were submitted for sequencing. Raw, paired-end sequence reads were processed using QIIME2 v. 2020.11 [21]. The DADA2 plugin was used to truncate reads at 250 bp and to denoise the paired-end reads prior to further analysis [22]. Taxonomy was assigned in QIIME2 using the Silva version 138.1, 99% Amplicon Sequence Variants (ASVs) from the 515F/806R classifier [23]. We then filtered the sequencing results for potential contaminants, or microbial taxa spuriously introduced during extraction, library

preparation, or sequencing, We defined contaminants as any bacterial taxa found predominantly in low biomass negative controls (at > 3% relative abundance), and found at relatively low proportions (< 3%) in true samples. These taxa were then bioinformatically removed from our data set (S1 Table). Unassigned ASVs were also bioinformatically removed. Sequences classified as chloroplasts were retained due to evidence that green algae cells and algal DNA may be found normally in developing amphibian larvae and adults respectively [24]. Samples with fewer than 1800 reads were also removed from analyses. This resulted in the retention of 90 skin swab samples and 2 water samples for further analysis.

We first analyzed the diversity and composition of the skin microbiota of all hellbenders (n = 90) by weight class (Low, Mid, High). Microbial diversity (alpha diversity: Shannon, Observed Features) and composition (beta diversity: Weighted and Unweighted UniFrac) were analyzed in QIIME 2. Microbial composition was compared across groups by weight class and tank using one-way permutational multivariate analysis of variances (PERMANOVA) employing weighted and unweighted UniFrac distance matrices. P-values were corrected for multiple comparisons using the Benjamini-Hochberg FDR correction, and values less than 0.05 were considered significant. An analysis of composition of microbes (ANCOM) on skin swabs was used to identify differentially abundant taxa by weight class. We filtered out taxa that had fewer than 10 reads and occurred in fewer than two samples prior to ANCOM analysis. We performed ANCOMs at the amplicon sequence variant (ASV) level, which is roughly equivalent to a bacterial strain level, however, deeper genome sequencing is necessary for true species and strain differentiation. Taxa that were identified as differentially abundant by weight class on skin swabs were noted as present or absent in the 2 water samples tested from tanks AQ-R2-1A and MF-1A.

We then performed multivariate adonis tests in QIIME2 to compare the effects of weight class, tank, and their interaction on skin bacterial diversity and composition. Next, we analyzed the effects of tank alone. Microbial diversity, composition, and differentially abundant taxa assessments were analyzed as described above. In our analysis, we only included tanks that held three or more hellbenders. All procedures were approved by the Ohio State University Institutional Animal Care and Use Committee (protocol 2020A00000051) and CZA Animal Use Committee.

## Results

### Skin microbial diversity and composition by weight class

There were no significant differences in skin microbial diversity of hellbenders (n = 90) by weight class (Low, Mid, High) (Shannon Diversity Indices $p = 0.598$, Observed Features $p = 0.159$, Fig 2). In contrast, there were significant differences in

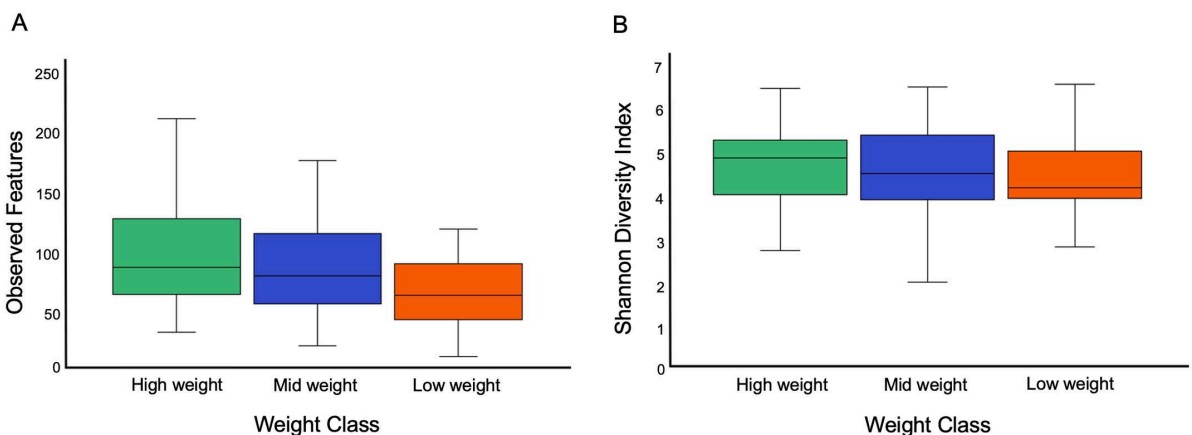

**Fig 2. Overall hellbender skin microbial diversity by weight class.** There were no significant differences in (a) Shannon Diversity Indices ($p = 0.598$) or (b) observed features among animals in different weight classes ($p = 0.159$).

microbial composition by weight class (PERMANOVA: Unweighted UniFrac $p=0.001$, pseudo-F$=2.06$; Weighted UniFrac $p=0.001$, pseudo-F$=2.87$; Fig 3). The skin microbiota in Low and Mid weight hellbenders were more similar to each other than to High weight hellbenders (PERMANOVA: Low to Mid, unweighted UniFrac $p=0.341$, weighted UniFrac $p=0.571$; Low to High, unweighted UniFrac $p=0.0015$, weighted UniFrac $p=0.003$; and Mid to High, unweighted UniFrac $p=0.0015$, weighted UniFrac $p=0.003$).

Five taxa were differentially abundant by weight class at the ASV level (Fig 4). Four taxa were increased in Low and Mid weight classes compared to the High weight class. Two of these taxa were especially increased in Low and Mid weight hellbenders including one taxa from family Comamonadaceae (ANCOM W$=1108$) and one from the genus *Nitrosospira* (ANCOM W$=1018$). Two additional differentially abundant taxa were particularly increased in Low weight hellbenders: one from the genus *Nitrospira* (ANCOM W$=1097$), and one from the genus *Parcubacteria* (ANCOM W$=1029$). A single taxa, *Pseudomonas peli*, was differentially increased in the High weight classes (ANCOM W$=1083$).

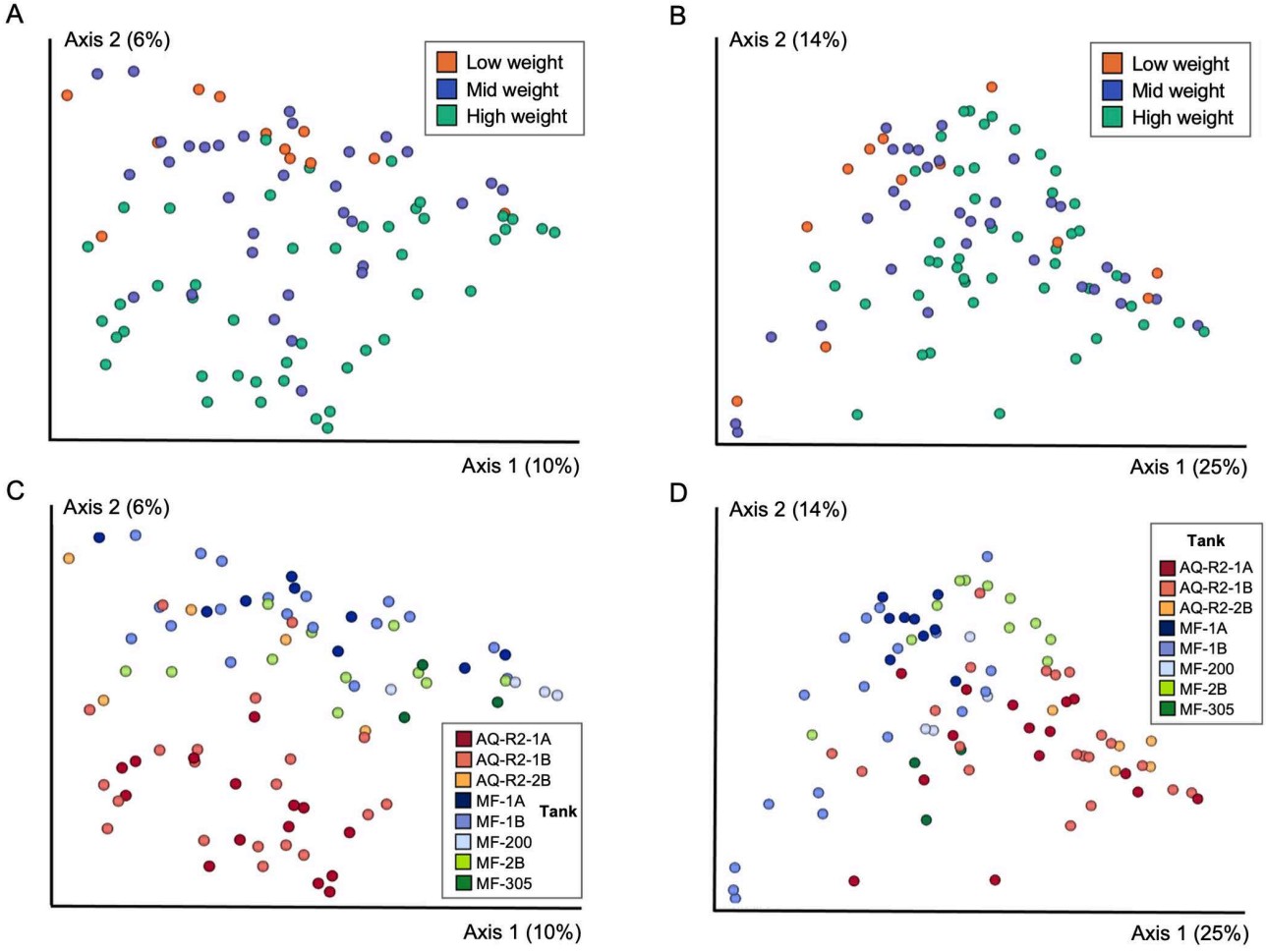

**Fig 3. Overall analysis of skin microbial composition by weight class.** Skin microbial composition differed significantly by weight class based on (a) unweighted and (b) weighted UniFrac metrics (n$=90$, PERMANOVA Unweighted $p=0.001$; Weighted $p=0.001$). Fig (c) and (d) represent the same distribution of points as in (a) and (b) but are colored by tank (n$=85$). Five samples were excluded from (c) and (d), as these hellbenders were in tanks with 2 or fewer individuals. A separate analysis was performed to evaluate microbial diversity and composition by tank. Hellbenders in tanks with 2 or fewer individuals were excluded from this analysis, which is shown in Fig 5. Each circle represents the bacterial community in a single hellbender.

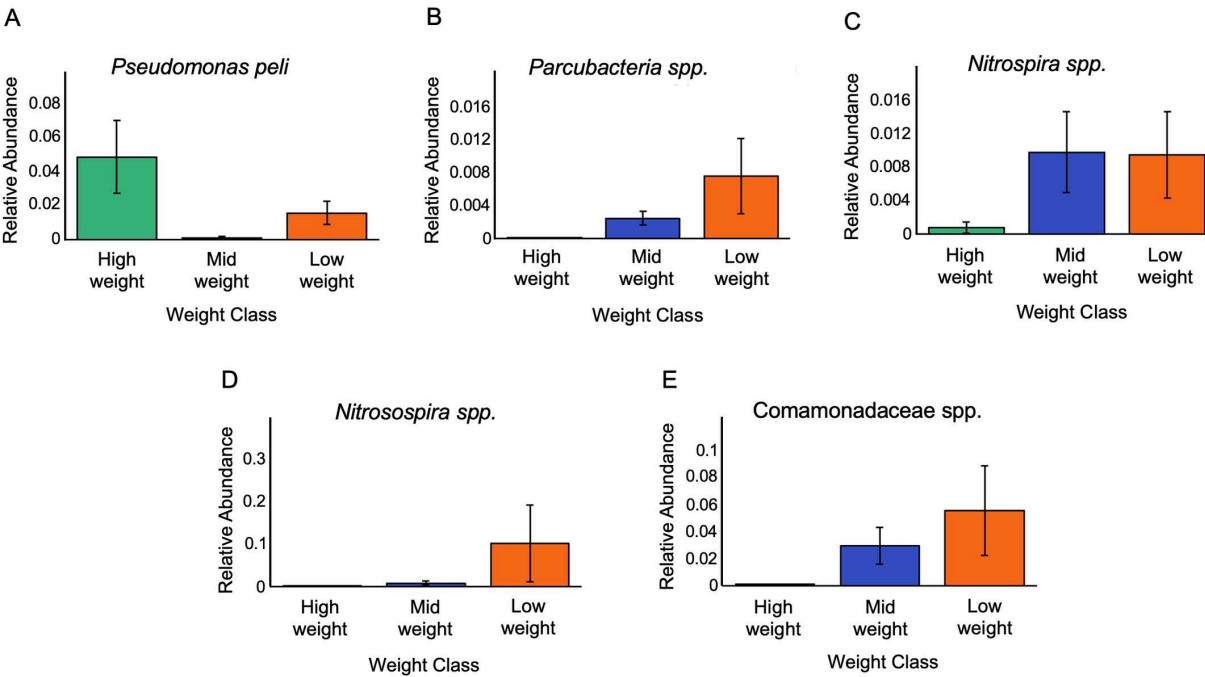

**Fig 4. Differentially abundant skin microbiota by weight class.** An ANCOM at the ASV level identified 5 differentially abundant taxa by weight class including (a) an ASV identified as a *Pseudomonas peli* species, (b) an ASV in the genus *Parcubacteria*, (c) an ASV in the genus *Nitrospira*, (d) an ASV in the genus *Nitrosospira*, and (e) an ASV in the Comamonadaceae family.

Of the five differentially abundant taxa, we found three (the Comamonadaceae and *Parcubacteria* taxa, and *Pseudomonas peli*) present in water (tanks AQ-R2-1A and MF-1A) (S1 Fig and S2 Table). The Comamonadaceae taxa displayed the greatest abundance in tank AQ-R2-1A, which contained primarily Mid and High weight animals (AQ-R2-1A = 1978 reads, relative abundance: 0.149; MF-1A = 37 reads, relative abundance: $1.89 \times 10^{-3}$). The two *Nitrospira* were not detected in either water sample, and *Pseudomonas peli* was only present in one water sample (tank AQ-R2-1A) and at low abundance (12 reads, relative abundance = $9.03 \times 10^{-4}$).

## Skin microbial diversity and composition by tank

As hellbenders were generally size sorted into tanks, we performed several additional analyses to parse potentially confounding effects between weight class and tank. First, we performed a multivariate adonis analysis that allowed us to account for both weight class and tank. The results showed that both weight class and tank had significant effects on skin microbial composition (Unweighted UnifFrac: weight class R = 0.21, $p < 0.001$; tank R = 0.44, $p < 0.001$; Weighted UniFrac: weight class R = 0.25, $p < 0.001$; tank R = 0.57, $p < 0.001$), with stronger tank effects observed compared to weight effects. There was also a significant interaction between weight class and tank based on unweighted UniFrac distances (R = 0.22, $p < 0.011$), but not weighted UniFrac distances (R = 0.20, $p < 0.11$).

We then reanalyzed microbial diversity and composition, but limited our analysis to eight tanks that each housed three or more animals (n = 85 hellbenders; eight tanks: AQ-R2-1A, AQ-R2-1B, AQ-R2-2B, MF-1A, MF-1B, MF-2B, MF-200, and MF-305). We found significant differences in overall diversity and composition by tank (Shannon: $p = 2.2 \times 10^{-4}$; PERMANOVA: Unweighted UniFrac $p < 0.001$, pseudo-F = 2.41; Weighted UniFrac $p < 0.001$, pseudo-F = 5.85) (Fig 5 and

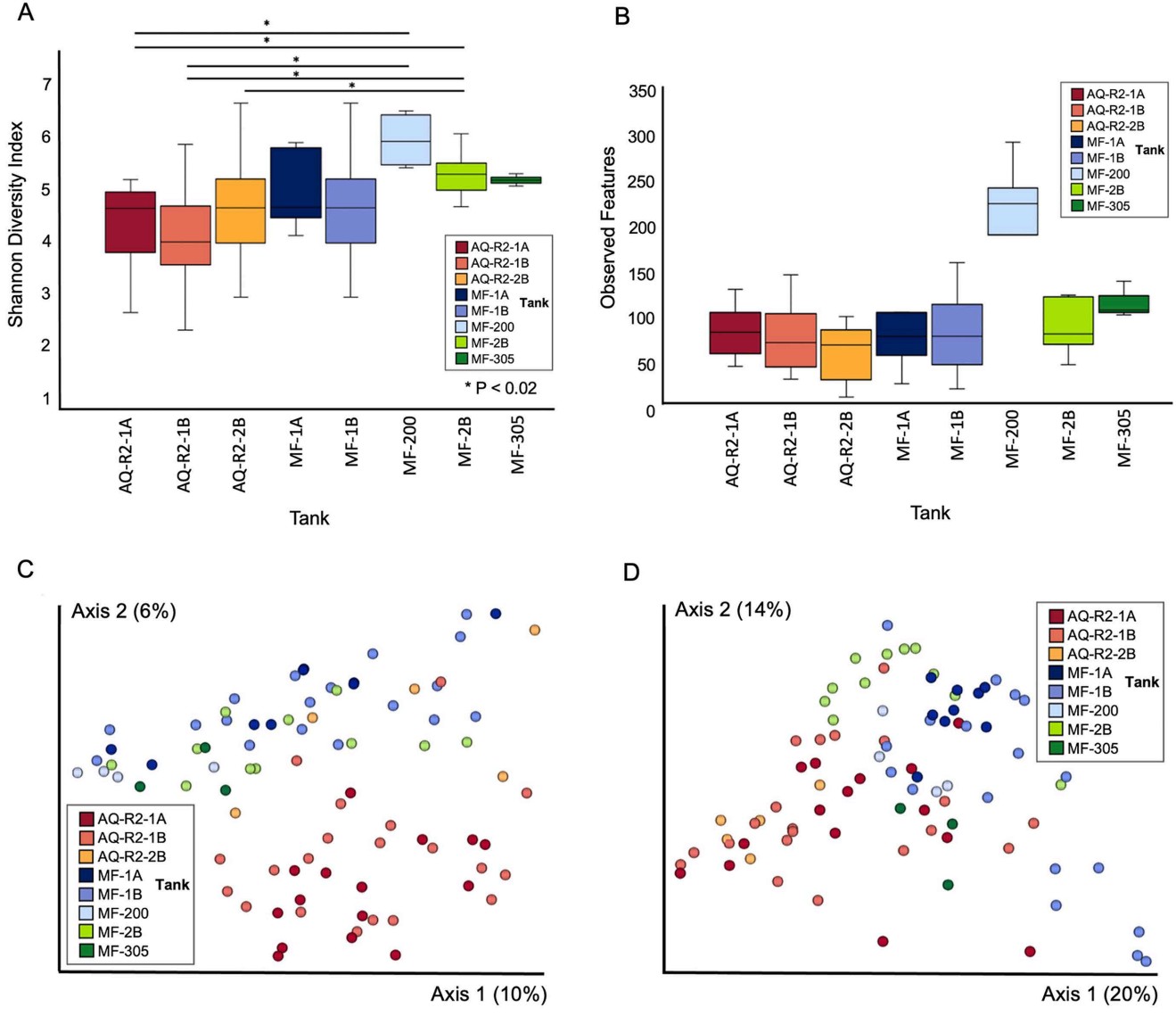

**Fig 5. Skin microbial diversity and composition of hellbenders by tank.** Microbial diversity differed significantly by tank based on the (a) Shannon Diversity Index (also see S3 Table) but not based on (b) Observed Features ($p = 0.085$). Skin microbial composition also differed significantly by tank based on (c) unweighted and (d) weighted UniFrac distance metrics. In (c) and (d) each circle represents a single hellbender and color represents the tank. Animals in tanks with 2 or fewer individuals were excluded from the analysis (n = 85 hellbenders retained in this analysis).

S3 and S4 Tables), with more pronounced differences found in tanks that were in separate rooms or buildings with different water and filtration systems.

We additionally identified 61 taxa (ASVs) that were differentially abundant based on tank (S5 Table). When we compared the five taxa that were differentially abundant by weight class to the 61 taxa that were differentially abundant by tank, we found that four taxa were present on both lists, but notably only one taxa (*Parcubacteria spp.*) was differentially abundant by weight class alone. This taxa (*Parcubacteria spp.*) was detected in animals from four unique tanks (AQ-R2-1B, AQ-R2-2B, MF-1A, and MF-1B).

## Discussion

In this study, we used culture-independent techniques to characterize the bacterial skin microbiota of Eastern hellbenders under human care, and we observed significant differences in animals among different weight classes as well as among animals housed in different tanks. Numerous host and environmental factors shape skin microbial communities in amphibians. Host immunity, for example, plays a key role in shaping the skin microbiota [12]. Skin mucus prevents evaporative water loss, creates a niche for various skin microbes, and also harbors antimicrobial peptides which are important for innate immune function [12,25]. Furthermore, skin infection with pathogenic microbes has been linked to reductions in amphibian skin microbial richness [12]. Differences in skin microbial richness have also been observed between grossly normal skin and chronic skin wounds in Ozark hellbenders (*Cryptobranchus alleganiensis bishopi*). Although it is unknown if the microbes detected in the skin wounds of this species are the primary cause of non-healing wounds or opportunistic infectious organisms, the presence of chronic skin wounds can cause long-term and significant negative health impacts in affected individuals [18].

We suspect that Eastern hellbenders classified as being in a Low weight in this study are a product of multiple factors within this population. Some individuals may be smaller as a result of normal variation as seen within any group. These smaller individuals likely experience greater levels of stress compared to larger individuals due to conspecific competition for space and food sources and in some instances suffer subsequent trauma. Chronic stress and impaired immune function may contribute to improper skin function, alterations in the skin microbiota, skin ulceration and necrosis, and poor wound healing [26]. Severe skin disease can lead to further systemic compromise, and ultimately result in significant morbidity and mortality in this species [27].

The significant differences we observed among animals housed in different tanks is consistent with previous research illustrating that the microhabitat of an animal is highly influential on the skin microbiome. Exposure to other animals, substrate, and water sources have been linked to significant patterns in amphibian skin microbiota [12]. Previous research characterizing the skin microbiome in free-ranging Eastern hellbenders has also shown differences in microbial community richness among different populations suggesting that local processes were an important influential factor [28]. Overall our findings are consistent with previous studies showing the importance of local conditions on the skin microbiota, however the majority of previous research has focused on free-ranging populations or comparing free-ranging and captive populations [29,30]. Our findings illustrate that fine-scale environmental differences in captive settings are linked to differences in skin microbial communities as well, and our results suggest these differences can be linked to skin and overall animal health.

Of the five differentially abundant taxa we identified based on weight class, two, the Comamonadaceae and *Parcubacteria* taxa, were present in water samples. This suggests that these taxa are potentially host-associated but shed easily into water; or they are water-associated but can also associate closely with hosts. The two Nitrospira taxa were not detected in either water sample, suggesting that Nitrospira is a host-associated taxa that is more firmly attached to its host or less well adapted to live freely in water. Similarly, *Pseudomonas peli* was only present in one water sample (tank AQ-R2-1A) and at relatively low abundances, but it was found at higher abundances on hosts, suggesting that it is a strongly host-associated taxa. Importantly, none of the differentially abundant taxa were detected in negative control samples, so they are unlikely to be contaminants introduced during the extraction and sequencing process.

Four of the five differentially abundant taxa based on weight class were also found to be differentially abundant based on tank. This result indicates that tank and weight class may be confounded. However, the *Parcubacteria spp.*, which was found in highest abundance on Low weight animals, was differentially abundant by weight class only and not differentially abundant by tank, suggesting a specific association between this taxa and Low weight animals. Moreover, the *Parcubacteria* taxa was detected in a water sample collected from tank AQ-R2-1A. None of the 25 animals in tank AQ-R2-1A (all Mid and High weight) contained *Parcubacteria* reads. However, the adjacent tank, AQ-R2-1B, which was connected on

the same water filtration system, contained Low, Mid, and High weight animals, and contained multiple hellbenders that produced skin swabs with *Parcubacteria* reads. Previous research shows that bacteria closely related to *Parcubacteria* are found widely in aquatic environments; however, the discordant results between hellbender skin swabs and water samples from tank AQ-R2-1A suggest that the presence or absence of Parcubacterium in hellbender skin in this population is not solely due to its presence in the surrounding aquatic environment.

*Parcubacteria* belong to a group of bacteria, the Candidate Phyla Radiation (CPR), which is a large group of uncultured bacteria [31,32]. *Parcubacteria* were first identified less than 25 years ago using rRNA sequencing techniques and since that time, CPR bacteria have been identified in numerous diverse aquatic environments including acidic, alkaline, marine, and freshwater systems. Amongst these diverse conditions, CPR bacteria have primarily been identified in limited oxygen or anoxic environments [33,34]. *Parcubacteria* are also distinctive from other bacteria in that they have a small cell size and a reduced genome. Many members of *Parcubacteria* have limited innate metabolic and respiratory capabilities and lack the ability to synthesize important compounds such as amino acids and fatty acids [34,35]. *Parcubacteria* also possess genes that code for glycosyltransferase enzymes which promote the biosynthesis of polysaccharides, polypeptides, and glycerolipids for essential metabolic functions including cell membrane synthesis and adhesion to host tissue [32]. These features suggest that many types of *Parcubacteria* rely on other organisms for essential metabolic functions and may act as parasites [35]. In hellbenders, this parasitic effect may be hindering optimal growth and contributing to increased metabolic stress. These cumulative effects can compromise systemic and skin health in Low weight animals. Ultimately these effects may contribute to the development of cutaneous abnormalities like those historically observed in captive-reared deceased juvenile hellbenders with low body condition and severe cutaneous lesions [16]. Additional studies on *Parcubacteria* are necessary and should evaluate its potential negative effect on hellbender growth and skin health. Moreover, further research is needed to determine if *Parcubacteria* may serve as a marker for skin dysbiosis in hellbenders within this system or in amphibians in other environments as well.

Amphibian skin and the associated microbiota are closely linked to an animal's overall systemic health. Alterations in habitat, environmental pollutants, and infectious pathogens have all been associated with significant changes in skin bacterial communities in amphibians [36]. Evaluating the impact of these factors on both free-ranging and captive amphibians is essential for understanding animal health on a population and individual level, and there is now an increased understanding that animals in captive rearing or breeding programs exhibit significant differences compared to their free-ranging counterparts [30,36]. Skin probiotic therapies have been used in various amphibian species and treatments have been linked to significant and physiologically important antifungal effects [8,12]. Infectious and non-infectious skin disease in amphibians poses a significant threat to both free-ranging and captive amphibian populations, and developing probiotic skin therapies is an important research priority for amphibian conservation [12,36].

There are several important limitations relevant to this study. First is that we focused on characterizing the bacterial microbiota in hellbenders and did not examine microbial eukaryotes such as fungi, which are an important factor in amphibian skin disease [37,38]. Secondly, serial longitudinal sampling of the study population could have helped identify baseline patterns and potential shifts in the microbiota over time. Lastly, the absence of functional assays in this study limits our ability to characterize and evaluate potential negative and positive microbial activities and their influence on hellbender health. Despite these limitations, we identified clear factors (weight class and tank) that were significantly associated with skin bacterial composition and we were able to identify a specific bacterial taxa, *Parcubacteria*, that may reflect or influence important changes in hellbender health and skin function. These findings would not have been possible using traditional culture-based techniques. Future research should prioritize how husbandry factors and potential stressors impact the skin microbiota in hellbenders and how these potential patterns are associated with morbidity and mortality. In addition, we suggest that management of hellbenders under human care may be improved by implementing several strategies. For example, baseline and subsequent serial evaluation of the skin microbiota may help better characterize normal shifts in the skin microbiota over time related to various factors including diet and environmental

conditions (e.g., water quality and tank properties). Serial testing can aid the identification of potential dysbiosis prior to the manifestation of obvious signs of disease. This may be especially helpful in this species which can exhibit long periods of cryptic and inactive behavior as part of their normal behaviors. With increased information about baseline and serial patterns of the skin microbiota in hellbenders, developing specific therapies (e.g., probiotic treatments) and management strategies to improve the health of hellbenders under human care would be invaluable to ex-situ conservation programs and ultimately free-ranging populations as well.

## Supporting information

**S1 Fig. Microbial relative abundances by phyla in water samples.**
(TIF)

**S1 Table. Taxa identified as contaminants.** A total of 13 taxa were identified as putative contaminants and bioinformatically removed from further analysis.
(TIF)

**S2 Table. Water samples.** Three of the five microbial taxa that were differentially abundant by weight class were also detected at variable abundances in water samples collected from tanks AQ R2-1A and MF-1A.
(TIF)

**S3 Table. Alpha-diversity pairwise comparisons of hellbender skin microbial diversity by tank.** Hellbender skin microbial diversity differed significantly by tank (Shannon Diversity Index: $p = 2.27 \times 10^{-4}$). Significant pairwise comparisons ($p < 0.05$, shaded in gray) revealed that hellbenders in tanks located in different buildings and maintained on different water systems were more likely to differ in microbial diversity than hellbenders located in tanks inside the same building/room or on a shared water system. Superscripts indicate tanks with a shared water and filtration system (shaded in yellow). These tanks were the most similar in terms of microbial diversity.
(TIF)

**S4 Table. Beta-diversity pairwise comparisons of hellbender skin microbial composition by tank.** Hellbender skin microbial diversity differed significantly by tank based on **(a)** unweighted and **(b)** weighted UniFrac distance metrics ($p < 0.001$). The majority of pairwise comparisons were significantly different ($p < 0.05$) indicating significant microbial composition differences by tank. Hellbenders in tanks that shared a water and filtration system (shaded in yellow) generally had more similar skin microbiota.
(TIF)

**S5 Table. Overall differentially abundant skin microbial taxa of by tank.** A total of 61 differentially abundant taxa (ANCOM) by tank were identified among tanks holding three or more hellbenders. Column labeled "Tank" indicates the tank in which the taxa was present in highest abundance. Column labeled "Wt Class" indicates if animals contained Low (L), Mid (M), and/or High (H) weight class animals.
(TIF)

## Acknowledgments

Assistance was provided by the Ohio State Department of Veterinary Preventive Medicine, the Columbus Zoo and Aquarium, the Ohio Hellbender Partnership, the Columbus Zoo and Aquarium Animal Health Department, the Ohio State/Columbus Zoo and Aquarium/The Wilds Zoo, Wildlife, and Ecosystem Health Residency Program, Greg Lipps (Ohio Hellbender Partnership), and John Navarro (Ohio Department of Natural Resources).

## Author contributions

**Conceptualization:** Andrea C. Aplasca, Randall E. Junge, Vanessa L. Hale, Mark Flint.

**Data curation:** Andrea C. Aplasca, Christopher Madden, Vanessa L. Hale, Mark Flint.

**Formal analysis:** Andrea C. Aplasca, Christopher Madden, Kilmer Soares, Vanessa L. Hale, Mark Flint.

**Funding acquisition:** Mark Flint.

**Investigation:** Andrea C. Aplasca, Randall E. Junge, Vanessa L. Hale.

**Methodology:** Andrea C. Aplasca, Peter B. Johantgen, Christopher Madden, Kilmer Soares, Randall E. Junge, Vanessa L. Hale, Mark Flint.

**Resources:** Peter B. Johantgen.

**Supervision:** Randall E. Junge, Vanessa L. Hale, Mark Flint.

**Writing – original draft:** Andrea C. Aplasca, Christopher Madden, Vanessa L. Hale, Mark Flint.

**Writing – review & editing:** Andrea C. Aplasca, Peter B. Johantgen, Christopher Madden, Kilmer Soares, Randall E. Junge, Vanessa L. Hale, Mark Flint.

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
