## [Decision Letter · Decision Letter 0]

2 Jun 2025

*Cryptobranchus alleganiensis alleganiensis*

Dear Dr. Hale,

Thank you for submitting your manuscript to PLOS ONE. After careful consideration, we feel that it has merit but does not fully meet PLOS ONE’s publication criteria as it currently stands. Therefore, we invite you to submit a revised version of the manuscript that addresses the points raised during the review process.

We look forward to receiving your revised manuscript.

Kind regards,

Alexandre Ribeiro da Silva

Academic Editor

PLOS ONE

Journal Requirements:

Reviewers' comments:

Reviewer's Responses to Questions

**Comments to the Author**

1. Is the manuscript technically sound, and do the data support the conclusions?

Reviewer #1: Yes

Reviewer #2: Yes

2. Has the statistical analysis been performed appropriately and rigorously?

Reviewer #1: Yes

Reviewer #2: Yes

3. Have the authors made all data underlying the findings in their manuscript fully available?

Reviewer #1: Yes

Reviewer #2: Yes

4. Is the manuscript presented in an intelligible fashion and written in standard English?

Reviewer #1: Yes

Reviewer #2: Yes

Reviewer #1: This is overall a neatly written paper with clear value to hellbender husbandry and reintroduction efforts. A few suggestions are as follows:

-The organization of the introduction seems like it would benefit from some tweaking. I would move the section about the importance of amphibian skin earlier in the paper to before hellbenders are introduced. That way you begin broadly (amphibians are important and declining, their skin and microbiota is an important part of their health) and then narrow down (hellbenders are a unique amphibian species of conservation concern, we're facing these challenges with reintroduction, skin may be the key)

-I would like more information about the eggs origins. How many unique clutches are they harvested from? Are they all from the same river reach? It seems that these could influence microbiome and overall health as well.

-Did you collect any data on the skin health of these guys? Given the discussions of lesions, I was curious if any of these individuals presented lesions and if you assessed differences in microbiome as it related to lesion presence or absence.

Line 84--This might better read "An organism’s habitat, including the soil, water, plants, and other animals to which they may be exposed, are important influences on a host."

Line 275--What software and packages were used for statistical analysis?

Line 313--This is a whopper of a sentence! I would break it into two.

Reviewer #2: PLOS ONE Reviewer Report

This manuscript meets PLOS ONE’s publication criteria with scientific rigor, clarity, and relevance. I recommend acceptance pending minor revisions.

This study offers valuable insights into the relationship between cutaneous microbiota, captive environmental conditions, and body condition in juvenile Eastern hellbenders—a species of high conservation priority.

Methodologically, the manuscript is robust, with a strong sample size (*n* = 116) and well-chosen analytical approaches (16S rRNA sequencing, PERMANOVA, ANCOM).

The findings have direct relevance for improving ex situ management practices. While the study is well executed, I recommend minor revisions to enhance clarity, statistical transparency, and ecological interpretation. The suggestions below aim to further strengthen the manuscript prior to publication. Notably, the identification of Parcubacteria as a weight-class–associated taxon presents a promising avenue for exploring host–microbiome dynamics in amphibians.

Recommendations for Improvement:

1.Abstract/Introduction:

•Replace "true association" (Line 55) with more precise language like "specific association". Better emphasize the practical applications for captive management

Statistical Analysis:

•PERMANOVA (Line 217): Clarify which factors were tested (e.g., weight × tank interaction) and report R² values for each factor in the results table.

•Multivariate Analysis: Consider testing whether tank effects persist after controlling for body condition.

2.Methods:

•Clarify criteria for weight class divisions (≤50g, 50-80g, ≥80g)

•Provide more details about tank maintenance protocols

•Expand description of contaminant removal process.

3.Results:

•Highlight that tank effects (R=0.44-0.57) were stronger than weight effects (R=0.21-0.25)

•Emphasize Parcubacteria as the only weight-specific táxon.

4.Discussion:

The observed association between Parcubacteria and low body condition in juvenile Eastern hellbenders suggests a potential role for this taxon in host health. Given its reduced genome and likely dependence on host-derived metabolites, Parcubacteria may act as a marker or contributor to microbial dysbiosis under physiological stress (Nelson & Stegen, 2015). Further discussion of its possible parasitic or opportunistic role would enhance the ecological interpretation of the findings.

- The manuscript should also acknowledge key limitations, such as the lack of longitudinal sampling to track microbiome shifts over time or in response to health outcomes, and the absence of functional assays to test microbial pathogenicity.

- To strengthen conservation applications, the authors are encouraged to propose practical management interventions. These include routine skin microbiome monitoring to identify early signs of dysbiosis, temporary segregation or enhanced care protocols for underweight individuals to limit microbial spread and stress, and pilot studies evaluating the probiotic potential of taxa like Pseudomonas peli, which was enriched in higher-weight animals.

- These additions would not only improve the biological depth of the study but also increase its value to captive amphibian husbandry and reintroduction programs.

Conclusion:

The manuscript represents a valuable contribution to amphibian conservation science. With minor revisions to improve clarity and emphasize practical applications, it will be suitable for publication in PLOS ONE. The findings have immediate relevance for hellbender captive breeding programs and may inform similar conservation efforts for other amphibian species.

**Do you want your identity to be public for this peer review?** For information about this choice, including consent withdrawal, please see our Privacy Policy

Reviewer #1: No

Reviewer #2: No

---

## [Author Response · Author response to Decision Letter 1]

29 Jul 2025

Response to reviewers document is included with this submission.

---

## [Decision Letter · Decision Letter 1]

21 Aug 2025

Environment and weight class linked to skin microbiome structure of juvenile Eastern hellbenders (*Cryptobranchus alleganiensis alleganiensis* ) in human care

PONE-D-25-05408R1

Dear Dr. Hale,

We’re pleased to inform you that your manuscript has been judged scientifically suitable for publication and will be formally accepted for publication once it meets all outstanding technical requirements.

Kind regards,

Alexandre Ribeiro da Silva

Academic Editor

PLOS ONE

Additional Editor Comments (optional):

Reviewers' comments:

Reviewer's Responses to Questions

**Comments to the Author**

Reviewer #1: All comments have been addressed

Reviewer #2: All comments have been addressed

2. Is the manuscript technically sound, and do the data support the conclusions?

Reviewer #1: Yes

Reviewer #2: Yes

3. Has the statistical analysis been performed appropriately and rigorously?

Reviewer #1: Yes

Reviewer #2: Yes

4. Have the authors made all data underlying the findings in their manuscript fully available?

Reviewer #1: Yes

Reviewer #2: Yes

5. Is the manuscript presented in an intelligible fashion and written in standard English?

Reviewer #1: Yes

Reviewer #2: Yes

Reviewer #1: (No Response)

Reviewer #2: The authors have satisfactorily addressed all prior concerns. Revisions to the Abstract/Introduction/Conclusion, improving interpretative accuracy. Additionally, the authors strengthened the manuscript by highlighting relevant examples of skin microbiome studies and management strategies in captive systems, thereby improving the practical conservation context of the study. Methods and Results sections have improved statistical clarity and methodological transparency. The expanded discussion on Parcubacteria provides a well-supported ecological interpretation, aligned with the study’s findings and broader conservation relevance. The manuscript now meets PLOS ONE’s criteria for scientific rigor, clarity, and data transparency. I consider it suitable for publication in its current form. I thank the authors for their careful revisions.

**Do you want your identity to be public for this peer review?** For information about this choice, including consent withdrawal, please see our Privacy Policy

Reviewer #1: No

Reviewer #2: No

---

## [Editor Report · Acceptance letter]

PONE-D-25-05408R1

PLOS ONE

Dear Dr. Hale,

I'm pleased to inform you that your manuscript has been deemed suitable for publication in PLOS ONE. Congratulations! Your manuscript is now being handed over to our production team.

Kind regards,

on behalf of

Dr. Alexandre Ribeiro da Silva

Academic Editor

PLOS ONE